# Role of *NF2* Mutation in the Development of Eleven Different Cancers

**DOI:** 10.3390/cancers17010064

**Published:** 2024-12-29

**Authors:** Shervin Hosseingholi Nouri, Vijay Nitturi, Elizabeth Ledbetter, Collin W. English, Sean Lau, Tiemo J. Klisch, Akash J. Patel

**Affiliations:** 1Department of Neurosurgery, Baylor College of Medicine, Houston, TX 77030, USA; shervin.hosseingholinouri@bcm.edu (S.H.N.); vijay.nitturi@bcm.edu (V.N.); elizabeth.ledbetter@bcm.edu (E.L.); collin.english@bcm.edu (C.W.E.); sean.lau@bcm.edu (S.L.); tiemo.klisch@bcm.edu (T.J.K.); 2Jan and Dan Duncan Neurological Research Institute, Texas Children’s Hospital, Houston, TX 77030, USA; 3Department of Otolaryngology–Head & Neck Surgery, Baylor College of Medicine, Houston, TX 77030, USA

**Keywords:** cancer, carcinoma, central nervous system (CNS), Hippo signaling pathway, Merlin, Neurofibromatosis 2, NF2

## Abstract

In this study, we sought to understand the role of NF2 gene mutation in the carcinogenesis of sporadic cancers. NF2 gene mutations are noted in several central nervous system tumors, solid-organ tumors, and skin cancers. We conducted a literature review on eleven different cancers with NF2 gene mutation involvement, summarizing the extent of association and specific biological pathways thought to be affected by NF2 mutations. We synthesized studies across several oncologic fields to consolidate what we know about NF2 gene mutations in cancer development. The Hippo signaling pathway is a biological pathway that is involved in eight of the eleven NF2-mutated cancers studied in this review. Although NF2 mutation has a known interaction with the Hippo signaling pathway, the specific details of this interface remain a topic for further studies.

## 1. Introduction

Cancer is an issue at the forefront of healthcare, with nearly two million Americans being diagnosed with some form of cancer each year. In 2024 alone, it is estimated that over 600,000 people will die of cancer in the United States [1]. Cancer is characterized by tumor formation due to unregulated cell proliferation and growth. Cancer can be regarded as a pathological ecosystem that harbors tumor microenvironment factors, like extracellular matrices and immune cells, in which neoplastic cells interact with these factors and themselves [2]. Tumorigenesis arises from somatic mutations in genes that are upstream cell pathway regulators or errors in genes that encode cell cycle proteins. Damage to the genome that leads to cancer formation can arise due to endogenous processes, like DNA replication, or exogenous factors, like UV radiation or chemical carcinogen exposure. Cancer therapies have diversified, with traditional treatment options being ineffective against many malignant tumors characterized by metastasis, heterogeneity, recurrence, and chemotherapy and radiotherapy resistance [3,4]. For the past thirty years, there has a been an uptrend in genomic analysis of various cancers, with an increase in use of genetic diagnostic tools like RNA sequencing and Whole-Exome Sequencing (WES). In parallel, there has been a four-fold increase in the number of gene therapy publications every year from 2000 to 2020. The increase in the number of genomic analyses of cancers and the parallel uptrend of genetic cancer therapy publications have helped provide more insights into key factors implicated in tumorigenesis, as well as molecular elements that can be targeted in cancer therapies.

Traditionally, carcinogenesis was thought to result from multiple subsequent clonal expansions driven by the accumulation of genetic alterations. However, with the increase in genomic cancer studies, it has become increasingly apparent that certain gene mutations have a much more nuanced impact on carcinogenesis. Some genetic mutations, known as driver mutations, confer a clonal advantage to cells and are positively selected-for in carcinogenesis. Most solid tumor malignancies are thought to require anywhere from 2 to 20 driver mutations that promote cell proliferation, thereby increasing susceptibility to further genetic aberrations over time [5]. Through analyses of the cancer genome, the Neurofibromatosis 2 (NF2) gene has been identified as one such gene that, when mutated, acts as a driver of mutations with the potential to promote the carcinogenesis of several cancers [6]. In other cancers, NF2 may play a less critical role and have an additive effect on the disease’s development. NF2 mutations have been implicated, to varying degrees, in several different cancer types, including sporadic central nervous system tumors, solid organ tumors, and tumors of the skin. In this review, we sought to elucidate NF2’s role in cancer formation and provide insights into the cellular pathways involved in NF2 mutant-driven carcinogenesis.

Neurofibromatosis 2 (NF2) is a gene located on the q arm of chromosome 22 which encodes for Merlin (moezin–ezrin–radixin-like) protein, a member of the protein 4.1 family of cytoskeletal elements [7]. Merlin consists of a globular amino-terminal FERM domain, a carboxy-terminal tail, and a flexible coiled-coil segment that joins the two ends. Merlin has a strong resemblance to ERM proteins that contain FERM domains, with the first 300 residues of Merlin sharing ~65% sequence identity with canonical ERMs. There are two common isoforms of Merlin: isoform 1, which has an extended carboxy-terminal tail encoded by exon 17; and isoform 2, which has an alternatively spliced exon 16 end. Merlin isoform 2 lacks the carboxy-terminal residues required for intramolecular binding with the FERM domain on the amino-terminal end of Merlin, leading to a constitutively active open conformation [8]. Merlin exists in several conformational forms, with a dephosphorylated active “closed” form and a phosphorylated inactive “open” form. In particular, Merlin is phosphorylated at Ser518 by p21 activated kinase (PAK1) or protein kinase A (PKA) and dephosphorylated by MYPT1-PP1 (Figure 1) [9]. Several signaling pathways crucial to cell proliferation are inhibited by the active Merlin protein, including PIKE-L/PI3K, mTORC1, Src/Fak, Mst1/2, Ras/Rac/PAK, ERK1/2, AKT, and CRL4-DCAF. In this way, NF2 acts as an essential tumor suppressor, inhibiting the development of several cancer types. NF2 is also thought to play a key role in cell–cell contact inhibition, although the role of localized accumulation of NF2 at the membrane level is unclear [7,10].

Within the q arm of chromosome 22 are adjacent genes to NF2 that are commonly co-mutated with NF2 in central nervous system tumors. Approximately 8-Mb away from NF2 and commonly involved in schwannoma tumorigenesis is the LZTR1 gene. According to the 4-hit/3-step model of schwannomagenesis, patients have a germline LZTR1 mutation that acts as the first hit, followed by a loss of heterozygosity (LOH) as the second hit. LOH is secondary to a mitotic recombination event or large deletion that encompasses the deletion of not only LZTR1 but also the neighboring NF2 allele. After this third mutational hit, a fourth hit occurs via an iatragenic NF2 mutation, leading to co-mutations of NF2 and LZTR1. The SMARCB1 gene is similarly located nearby NF2, on chromosome 22q, and is commonly co-mutated with NF2 following the four-hit/three-step model [11]. Studies have shown evidence for additional somatic mutations in NF2 in schwannomas characterized by germline mutations in LZTR1 or SMARCB1, further supporting evidence for the four-hit/three-step model of schwannomagenesis [12].

Germline mutations in NF2 lead to autosomal dominant neurofibromatosis type 2, commonly characterized by bilateral vestibular schwannomas, ependymomas, cranial meningiomas, and spinal nerve root schwannomas [13]. Neurofibromatosis type 2 disease has a prevalence of roughly 1 in 30,000 live births, with around 50% of cases having de novo mutations [6]. Most patients with neurofibromatosis type 2 initially present with vestibular schwannomas, followed by other central nervous system tumors, skin tumors, and ocular manifestations [14]. Whereas germline NF2 mutations lead to neurofibromatosis type 2, somatic mutations in the NF2 gene inhibit NF2’s tumor suppressive effects, thereby promoting the formation of various cancerous tumors [13,15]. More specifically, sporadic NF2 tumors are thought to follow Knudson’s two hit hypothesis, whereby the second hit event often involves the loss of all or most of chromosome 22 and, most frequently, LOH [7]. Sporadic NF2 mutations have been implicated in several cancers, including sporadic meningioma, ependymoma, and schwannoma, mesothelioma, breast cancer, hepatocellular carcinoma, prostate cancer, glioblastoma, thyroid cancer, melanoma, and renal cell carcinoma.

## 2. Materials and Methods

We reviewed the scientific literature for reputable evidence highlighting the involvement of NF2 mutations in the development of different cancers. For each cancer, we found several experimental studies providing evidence for NF2-mutant tumors. Studies included were those on *H. sapiens* cells and *M. musculus* cells and models. Furthermore, we elucidated the biological pathways involved in the carcinogenesis of these NF2-mutated tumors to better understand key trends in NF2-related cancers. We conducted a topical order review of cancers associated with NF2-mutations. We used the key words “NF2”, “Merlin” with “Cancer”, “Carcinoma”, and “Driver Mutation” to find cancers associated with mutations in the NF2 gene. Next, we conducted a general-to-specific review of the biological pathways involved in NF2-mutant cancers. Most review papers summarizing NF2 aberrations in cancers were excluded. We only used the most up-to-date research articles in each respective cancer field that outlined NF2 mutation involvement. We verified that all research articles used in this review were published in reputable peer-reviewed journals.

## 3. Results

### 3.1. Review Results

#### 3.1.1. Meningioma

Meningiomas are tumors arising from arachnoid cap cells and are the most common central nervous system tumors, accounting for over one-third of all primary intracranial tumors [15,16]. Around 20% of meningiomas are characterized as aggressive, with more frequent recurrences and an average 10-year survival of around 50% [15]. Molecular differences in meningiomas have different demographic characteristics, with NF2-mutant meningiomas being more common in males, while KLF4- and POLR2A-mutant meningiomas are more common in females [17]. Monosomy, NF2 mutations, and TRAF7 mutations seem to be the drivers of tumorigenesis in sporadic meningiomas, and as many as 60% of all sporadic meningiomas involve a loss of heterozygosity on chromosome 22 due to inactivating mutations of NF2 [15,18,19]. Meningiomas not defined by NF2 mutations commonly have deletions in CDKN2A/B located on chromosome 9p, as well as TRAF7 mutations on chromosome 16p [20,21]. Nearly a quarter of all meningiomas are characterized by TRAF7 loss, which is mutually exclusive with NF2 mutation [21]. NF2 has been implicated in several pathways that enable its role as a tumor suppressor. The inactivation, mutation, or loss of NF2 in meningiomas leads to loss of NF2 tumor suppressor activity, most notably by way of altered E-Cadherin expression, CD44-mediated cell–cell contact inhibition, and the Hippo signaling pathway.

Inherent to Merlin’s growth inhibitory function is its induction by intracellular adhesion and its inactivation by integrin signaling. Cadherin engagement at the membrane level inactivates PAK, resulting in an accumulation of dephosphorylated active Merlin. At the same time, integrin attachment to the extracellular matrix (ECM) activates PAK, highlighting the independent regulation of Merlin from contact-mediated signaling events [8]. Through analysis of the differential expression of E-Cadherin across the different World Health Organization (WHO) histopathological subgroups of meningiomas, it was found that aggressive WHO II meningiomas involve a loss of E-Cadherin and related Zo-1 compared to WHO I meningiomas. Aggressive WHO II meningiomas also have increased epithelial to mesenchymal transition (EMT) relative to WHO I meningiomas. EMT is a complex biochemical process that results in the loss of polarity of the plasma membrane and a subsequent increase in the invasive and migratory properties of the membrane. EMT has been associated with increased invasiveness of epithelial tumor cells. Several studies have shown histological evidence of EMT in other epithelial cell cancers, like nasopharyngeal carcinoma (NPC) and malignant mesothelioma (MM), which is thought to contribute to neoplastic spindle cell development in NPC and aggressive cancer phenotypes in MM [22,23]. More specifically, aggressive meningiomas with allelic losses of NF2 show higher levels of several EMT-relevant proteins [24]. Therefore, a proposed mechanism for the loss of tumor suppressor activity in NF2-mutated sporadic meningiomas is through a loss of E-Cadherin activity.

Cell–cell contact inhibition is another way in which NF2 regulates cell activity as a tumor suppressor. Contact inhibition is mediated by the Merlin protein through its interaction with CD44, an ECM component that is a cell–cell adhesion molecule hyaluronan. Merlin binds to the cytoplasmic tail of CD44 in its unphosphorylated active form, thereby inhibiting CD44 and facilitating contact inhibition at high cell densities. This Merlin-mediated contact inhibition is regulated by the recruitment of RAC1 GTPase to the membrane, which inhibits the activation of RAC1 and PAK1 that would otherwise phosphorylate and inactivate Merlin [9,25]. With somatic mutations of NF2, Merlin loses its tumor suppressive regulation of cell–cell contact inhibition through CD44, contributing to the formation of sporadic meningiomas.

Perhaps the most extensively researched mechanistic pathway relevant to the tumor suppressor activity of the NF2 gene in sporadic meningiomas, as well as other cancers, is the Hippo signaling pathway. The Hippo signaling pathway is a highly conserved kinase cascade initially discovered in drosophila. The Hippo pathway plays a key role in the regulation of target genes responsible for tissue homeostasis, organ size, cellular proliferation, cellular survival, cellular differentiation, and stem cell behavior [8,26]. Central to the canonical Hippo signaling pathway are the MST1/2 and MAP4K/TAOK kinases in parallel, which are directly phosphorylated by LATS1/2 kinases. LATS1/2 kinases, along with adaptor proteins SAV1 and MOB1A/B, phosphorylate and inhibit downstream YAP1 and its paralog TAZ. Phosphorylated YAP1 gets sequestered to the cytoplasm and binds to 14-3-3 proteins, leading to the degradation of YAP1 in the Hippo ‘on’ state. In the Hippo ‘off’ state, the aforementioned kinase cascade does not phosphorylate YAP1, allowing YAP1 to localize in the nucleus, where it interacts with TEAD1-4 transcription factors that promote gene expression. Merlin is thought to act as an upstream regulator of the Hippo signaling pathway, although the precise interaction partners within the pathway are not known. It is speculated that Merlin localizes in the cell membrane, where it recruits and activates MST1/2, LATS1/2, or both, leading to the Hippo ‘on’ state. KIBRA is thought to serve as a scaffold complex that localizes these kinases in the cell membrane for subsequent activation. With unrestrained YAP/TAZ activity in the absence of wildtype Merlin, unrestrained cell growth through Hippo gene expression leads to tumorigenesis. Similarly, complete NF2 deletion rescues Hippo ‘off’ state activity, leading to tumor progression. Although NF2 plays an established role in Hippo signaling activity regulation, the specific role of Merlin within the Hippo signaling pathway and the differentiation of NF2 mutant activity within the Hippo signaling pathway remain unknown.

Merlin is a known plasma membrane-associated protein required for the activation of the Hippo signaling cascade, thereby effecting the activity of LATS1/2. In addition to the effect that LATS1/2 has on downstream Hippo signaling targets YAP and TAZ, LATS1/2 phosphorylation leads to the inhibition of MTF1, a transcription factor essential to the heavy metal response. With NF2 mutations, LATS1/2 is downregulated, leading to a lack of inhibition of MTF1, which results in the upregulation of heavy metal response gene transcription and cellular protection. This was evident in NF2 KO cells that harbored resistance to Cd/Zn-induced toxicity [27]. LATS1/2 activity is also linked with the regulation of interferons (IFNs), which have broad-spectrum antiviral activity. Upon LATS1/2 phosphorylation in the Hippo signaling cascade, phospho-LATS1/2 directly promotes IFN-mediated antiproliferation. Additionally, activated LATS1/2 phosphorylates STAT1, inducing full IFN-I antiviral activity. With a deficiency in LATS1/2, it has been shown that in vivo IFN-I signaling is restricted, bringing about dysregulation of host antiviral immune response [28]. NF2 mutations lead to the inactivation of LATS1/2, thereby reducing the activity of STAT1 and attenuating IFN-I signaling activity.

#### 3.1.2. Other CNS Tumors

A loss of Merlin expression is characteristic of sporadic ependymomas and schwannomas, in addition to meningiomas. Many of these sporadic CNS tumors show considerable reduction or deletion of Merlin expression. Furthermore, sporadic schwannomas like meningiomas oftentimes have nonsense mutations that result in truncated Merlin proteins [29]. The resulting truncated Merlin proteins in these tissues inhibit normal NF2 tumor suppressive functions, contributing to tumorigenesis. Notably, a significant fraction of sporadic ependymomas with spinal localization have chromosome 22 aberrations and NF2 mutations [30]. An NF2 gene panel of over 200 schwannomas, including NF2-schwannomas, sporadic schwannomas, and schwannomatosis-related schwannomas, found that 79% of these tumors had a second mutational event that causes sporadic tumor formation. Most commonly, the second hit event is LOH. Furthermore, a gene panel study found that around 66% of all sporadic vestibular schwannomas had a somatic NF2 mutation [31]. Specifically in sporadic vestibular schwannomas, NF2 is the central factor in tumorigenesis, with anywhere from 15 to 84% of tumors harboring NF2 mutations [32]. Therefore, NF2 mutations at the somatic level account for the tumorigenesis of over half of all sporadic meningiomas, ependymomas, and schwannomas.

The key regulatory pathway thought to be involved in NF2-mutant sporadic schwannomas is the Hippo signaling pathway. NF2 is thought to act as an upstream regulator of the LATS1/2 kinases in the Hippo signaling cascade. With NF2 mutations, LATS1/2 is thought to be inhibited, thereby not regulating the downstream YAP protein, which localizes in the nucleus for the upregulation of cellular survival and growth gene expression. A sequencing study of 82 human peripheral and spinal schwannoma samples found that 45 cases (55%) had NF2 mutations. The same study looked at promoter methylation of LATS1/2, as well as nuclear YAP expression and cytoplasmic phosphorylated-YAP. The study found that there was LATS1 promoter methylation in 14 cases (17%), LATS2 promoter methylation in 25 cases (30%), nuclear YAP expression in 18 of 42 cases (43%), and reduced cytoplasmic phosphor-YAP expression in 15 of 49 cases (31%). Ultimately, the active Hippo signaling pathway converges on YAP nuclear localization and reduces cytoplasmic phospho-YAP expression, whereby increased values of either indicate dysregulation in one of the upstream genes, including NF2, LATS1, and LATS2. The values from this sequencing study indicated that NF2-mutant schwannomas had reduced regulation of the Hippo signaling pathway, promoting cell proliferation and the pathogenesis of sporadic schwannomas [33]. A newer study mirrored these results, showing regression of schwannoma growth after genetic ablation of YAP and TAZ or after the application of TEAD palmitoylation inhibitors in mouse schwannoma models [34].

#### 3.1.3. Mesothelioma

Mesothelioma tumors are amongst the other non-CNS solid tumors that can result from somatic mutations in the NF2 gene. Interestingly, an analysis of 75 different human lung cancer cell lines, including 38 small cell, 34 non-small cell, and 3 carcinoid cell lines, found no NF2 mutants. NF2 mutations were, however, found in mesotheliomas as either nonsense truncating mutations or large deletions. Moreover, a significant fraction of mesothelioma tumors have cytogenetic abnormalities in chromosome 22, where NF2 is located [35]. Around 30% of malignant mesotheliomas have been identified with somatic mutations in NF2. NF2 mutants’ loss of tumor suppressor activity is thought to be related to the Hippo signaling pathway activity within these tumor tissues. Targeted therapeutic trials are underway for malignant mesothelioma in the form of TEAD palmitoylation inhibitors that block the proliferation of NF2-deficient tumors by blocking Hippo signaling gene expression [36]. NF2-mutations as they relate to Hippo signaling pathway regulation have been studied in human malignant pleural mesothelioma (MPM) cells as well. These MPM tumor cells highlighted a different pathway involved in the tumorigenesis of NF2-mutant mesothelioma. NF2-mutant MPM cells were not associated with changes in phospho-YAP levels or YAP/TAZ bioactivity. Rather, these tumors were characterized by a deficiency in the B-cell receptor signaling pathway (BCR) [37]. Although no consensus has been made on the specific biological pathway involved in NF2-mutant mesothelioma tumors, it is clear that NF2 mutations are characteristic of these tumors. Further studies must be conducted to elucidate the specific pathway involved in cancer formation in NF2-mutant mesotheliomas, be it the Hippo signaling cascade, BCR, neither, or both.

#### 3.1.4. Breast Cancers

Although low in prevalence, somatic NF2 mutations are also characteristic of some breast cancers. However, several studies have found frequent LOH at the loci on the q arm of chromosome 22 in breast cancers. More recently, a study looking at NF2 expression in breast cancer cells found that there was relative downregulation of NF2 in breast cancer tissue compared to non-neoplastic adjacent tissue [38]. Merlin protein expression is noted to be significantly reduced in metastatic breast cancer tissues. Using an NF2 knockout mouse mammary tumor model, a study showed that these NF2-silenced cells had reduced activity of the nuclear factor erythroid 2, like 2 antioxidant transcription factor and increased expression of NADPH oxidases. The characteristic malignancy of these breast cancer tissues is thought to be related to the dysregulation of cellular redox management in these NF2 mutant cells [39]. Another study found that BRCA1 and BARD1 inhibit NF2 through ubiquitination, thereby reducing Hippo signaling-mediated cell proliferation and tumorigenesis. Therefore, a lack of NF2 ubiquitination in BRCA1- and BARD1-deficient breast cancer cells contributes to tumor formation [40]. Triple-negative breast cancer (TNBC) has a particularly poor prognosis due to a lack of target treatment plans, but recent studies using TNBC human cells have provided some insightful discoveries. Most of these TNBC cells are characterized by elevated levels of CD24, making them insensitive to paclitaxel chemotherapeutics but sensitive to ferroptosis agonists. Specifically, TNBC cells were found to be sensitive to modulated NF2-YAP signaling, altering the levels of ferroptosis suppressor protein 1 (FSP1) and CD24. When NF2 was inhibited and YAP was overexpressed in TNBC cells, CD24 and FSP1 were inhibited, resulting in enhanced ferroptosis and TAM phagocytosis, hindering TNBC tumor growth [41]. Therefore, the role of NF2 mutations is more nuanced in breast cancer, as NF2 loss by way of the Hippo signaling pathway can both contribute to tumorigenesis in some cases but also inhibition of tumor growth in TNBC tumors.

#### 3.1.5. Hepatocellular Carcinoma

The implications of NF2/Merlin loss in liver cancer models have also been explored. A study from 2010 created liver specific NF2 mutations using a mouse model. All mice with NF2 mutations in this study eventually developed both hepatocellular carcinoma and cholangiocellular carcinoma. The same study also delved into the underlying mechanism promoting tumorigenesis in the liver without NF2. They found that in the liver, Merlin is not a major regulator of YAP in the Hippo signaling cascade. Rather, it is believed that the overexpression of NF2−/− is driven by aberrant EGFR activity, and either that additional mutations cooperate with Nf2 loss to drive the development of HCC or HCC develops from Nf2−/− progenitors that adopt an improperly differentiated state [42]. Another study investigated the effect of NF2 mutations on primary liver cancers in human liver tissue. The study looked at NF2 mutations as well as a theorizing a downstream HIPPO signaling pathway gene, i.e., YAP1. They found that NF2/Merlin was expressed in higher levels in the tumor tissue of HCC compared to adjacent non-tumor tissue but was significantly less expressed in the tissue of intrahepatic cholangiocarcinoma (ICC) compared to adjacent non-tumor tissue. Furthermore, the study found that there was a negative correlation between Merlin and YAP expression in ICC tissue, indicating the role of the HIPPO signaling pathway in the development of liver carcinomas in NF2 altered liver tissue [43]. Thus, it is unclear whether NF2-mutant liver tissue transforms into HCC by way of altered regulation of the Hippo signaling pathway or via another biological pathway, like the EGFR pathway.

#### 3.1.6. Prostate Carcinoma

The second most common cause of cancer mortality in the male US population is prostate cancer, which has associations with mutations in the NF2 gene as well. A study investigating several prostate cancer cell lines revealed that NF2 expression was significantly lowered in these cells. More specifically, the NF2 activity in these prostate carcinoma cell lines were suppressed by the upregulation of p-21 activating kinases (PAKs) that constitutionally phosphorylate NF2. Upon phosphorylation, NF2 assumes its “open” conformation, inactivating its tumor suppressor activity. Besides NF2, the same study found that several other tumor suppressors, including OPCML, Cav-1, and UGT2B17, showed a loss of tumor suppressive effects. The mechanism of this tumor suppressor loss is unknown, but it was theorized that it may be related to genetic or epigenetic modifications. These modifications are thought to progress indolent prostate cancers to more proliferative and aggressive carcinomas in an additive manner [44]. Another study looked at the phosphorylation of NF2 at Ser518 by PAKs, which inactivates the tumor suppressive effects of NF2. Western blots against anti-phospho-Merlin (Ser518) were examined across the prostate cancer cell lines LNCaP, DU145, PC3, 22RV1, and LAPC-4. In the LNCaP, PC3, 22RV1, and LAPC-4 prostate cancer cell lines, Merlin expression as well as phosphor-Merlin were low, whereas DU145 cells had high expression of NF2 but also PAK activation. Therefore, it was concluded that Merlin was inactivated in DU145 prostate cancer cells by PAK-mediated inhibition, further providing evidence for the role of NF2 in prostate cancer development [45]. Although the role of NF2 downregulation in prostate carcinoma is well-defined, the specific pathways involved in NF2 inhibition have yet to be explored and should be a focus in future studies.

#### 3.1.7. Glioblastoma

The third most common brain tumor and the most aggressive are glioblastomas, which have median survival of roughly 15 months. Although glioblastomas are not associated with NF2 mutations directly, several studies using glioblastoma cell lines have uncovered the role of Merlin phosphorylation in glioblastoma development. One study found that select sub-populations of U251 glioblastoma cells were characterized by high expression of phosphorylation of Merlin at Ser518. These same subpopulations also had high expression of NOTCH1 and epidermal growth factor receptor (EGFR), which with phosphor-Merlin correlated to increased cell proliferation and tumorigenesis. More specifically, increased expression of phospho-Merlin was found to inhibit the cell–cell contact inhibition related to Merlin’s interactions with HES1 and CCND1, thereby further promoting tumorigenesis in these cells [46]. Another study of glioblastomas found that NF2 was lost in nearly a third of all glioblastoma cell lines and tumors. Interestingly, the study looked at the interaction between Merlin and Ezrin, with the latter being from the same family of proteins as Merlin. They found that Ezrin interacts and releases NF2 from the cortical compartment, reversing NF2-mediated inhibition of Rac1. With glioblastoma tissue that has an overexpression of both NF2 and Ezrin, Ezrin can inhibit Rac1 inhibition, leading to proliferation. With the overexpression of only Ezrin, this increase in tumorigenesis and cell proliferation is not seen [47]. Although commonly mutated and involved in several pathways that promote tumor development, NF2 does not seem to play an essential role in the tumorigenesis of glioblastomas. Rather, NF2 mutations may help contribute to poorer phenotypes of glioblastomas, characterized by key mutations in other driver mutations.

#### 3.1.8. Thyroid Cancers

Another non-CNS cancer that has NF2 implications is thyroid cancer. Medullary thyroid carcinoma, which accounts for less than 10% of all thyroid malignancies, is associated with NF2 loss. A study that analyzed 11 cases of medullary thyroid carcinoma looked for allelic losses of known tumor suppressor genes. The study found that the most frequent allelic losses were in NF2, followed by L-myc and p53. Allelic loss of NF2 was noted in 75% of the 11 cases analyzed in the study, whereas L-myc and p53 allelic loss was noted in 44% [48]. Additionally, NF2 loss in combination with HRAS mutation leads to murine poorly differentiated thyroid cancer (PDTC). The loss of NF2 in these tumors results in RAS signaling, partially due to the inactivation of the Hippo signaling pathway. With the loss of NF2, the Hippo signaling pathway kinase cascade is inhibited, leading to YAP localization in the nucleus. Within the nucleus, YAP interacts with TEAD1-4, leading to the upregulation of gene expression related to cell proliferation. Specifically, the study on PDTC found that RAS genes are upregulated by YAP-TEAD1, leading to RAS-induced tumorigenesis in NF2 absent tissue. The study confirmed YAP-TEAD involvement in RAS-induced tumorigenesis by inhibiting cell growth through verteporfin, a compound that disrupts YAP-TEAD transcription [49]. The specific biological pathway involved in human NF2-mutant thyroid cancers is a subject for further study, but current evidence provides support for Hippo signaling and Ras signaling involvement.

#### 3.1.9. Melanoma

In terms of skin carcinomas, melanoma has been verified through several studies to have a link to NF2. A study from 2012 found that there was increased expression of Merlin in certain metastatic melanoma cell lines. This increase in Merlin expression reduced the cell lines’ in vitro migration and proliferation while also inhibiting the in vivo growth of melanoma cells in mice. Additionally, the study found that the overexpression of NF2 promotes hydrogen peroxide-induced activation of MST1/2, linking NF2 to the Hippo signaling cascade in melanoma cells. Merlin knockdown in WM1552C human melanoma cells promotes subcutaneous growth, further supporting NF2’s role in melanoma tumorigenesis [50]. Another study found implications for NF2 by looking at NF2-mutant tissues’ effect on the cGAS-STING signaling pathway, a pathway involved in initiating antitumor immunity. Using murine B16-F10 melanoma cells, they found that NF2s with mutations in the FERM domain were potent suppressors of cGAS-STING signaling. These mutant NF2s tightly associate with IRF3 and TBK1, which then create cellular condensates with activated IRF3 that eliminate TBK1 activation [51]. Therefore, within melanoma studies, there exists conflicting evidence for the specific pathway that NF2-mutant cells are involved in regarding their role in cancer formation.

#### 3.1.10. Renal Cell Carcinoma

Renal cell carcinoma (RCC) is the most common malignancy of the kidneys and is further subclassified into clear cell RCC (ccRCC, 75%), non-clear cell RCC (nccRCC, 25%), and other exceedingly rare subtypes, like medullary and collecting duct RCCs. Molecular profiles of RCC have been largely expanded on, with ccRCCs commonly having mutations in VHL and nccRCC; these are associated with mutations in genes like FH, cMET, SDH, and SMARCB1, to name a few. A study that performed a comprehensive genomic profiling (CGP) of over 3900 clinically advanced kidney tumors found that 192 (4.9%) RCC tumors had NF2 mutations. More specifically, 15–30% of all nccRCC had NF2-mutations, confirming the role of NF2 in the tumorigenesis of nccRCC [52]. The specific mechanism by which NF2 mutations induce tumorigenesis in RCC is thought to be via the Hippo signaling pathway. A study using an inducible orthotopic kidney tumor model found that YAP/TAZ silencing induces regression of established NF2-mutant tumors, prolonging survival in mice with NF2-mutant RCC. This study provided evidence for the role of the Hippo signaling pathway in tumorigenesis of NF2-mutant kidney tissues [53]. Another study with human sarcomatoid nccRCC tumor specimens reiterated the involvement of the Hippo signaling pathway in NF2-mutant RCC tumors. Through immunohistochemistry studies, those authors found that NF2-mutant sarcomatoid nccRCC had upregulation of YAP/TAZ nuclear protein. With YAP1 knockdown and NF2 reconstitution, they found that tumor growth and proliferation was inhibited [54]. Through these studies, it is evident that NF2 mutations are characteristic of nccRCC, and particularly, that dysregulation in Hippo signaling pathway activity leads to tumorigenesis in kidney tissue.

## 4. Discussion

NF2 gene mutations have been identified in several cancers, many of which have implications in the Hippo signaling pathway. NF2 is a known tumor suppressor which, under non-cancerous conditions, acts to regulate cell proliferation and growth. One of the key pathways in which NF2 is thought to play a role as a tumor suppressor is the Hippo signaling pathway. The Hippo signaling pathway is an evolutionarily conserved kinase cascade originally discovered in drosophila that regulates target genes which are responsible for tissue homeostasis, organ size, cellular proliferation, cellular survival, cellular differentiation, and stem cell behavior. Central to the Hippo signaling pathway is a series of kinases that starts with MST1/2 (Figure 2). When MST1/2 is phosphorylated by upstream regulators, it becomes activated and phosphorylated LATS1/2. LATS1/2 are also activated by phosphorylation and go onto phosphorylate YAP/TAZ. The phosphorylation of YAP/TAZ ensures that YAP/TAZ is retained in the cytoplasm, where it interacts with 14-3-3 proteins and is eventually degraded. When upstream regulators do not phosphorylate MST1/2, LATS1/2 is not phosphorylated, which enables unphosphorylated YAP/TAZ to localize in the nucleus. In the nucleus, YAP/TAZ interacts with transcription factors TEAD1-4, upregulating the gene expression associated with cell growth and proliferation. This state of Hippo signaling is referred to as the Hippo “off” state, where a lack of activity in the Hippo kinase cascade leads to eventual gene expression related to cell proliferation.

Under normal conditions, the NF2 gene encodes for the Merlin protein, which acts as an upstream regulator of the Hippo signaling pathway. NF2 is thought to interact with MST1/2, LATS1/2, or a combination of the two, leading to their phosphorylation. As previously discussed, activation of the Hippo signaling kinase cascade leads to the phosphorylation of YAP/TAZ and its eventual degradation, inhibiting the interactions with YAP/TAZ and TEADS1-4 that would upregulate gene expression. In this way, NF2 acts as a tumor suppressor through the Hippo signaling cascade by downregulating cell proliferation and survival, hallmark characteristics of all cancers. When NF2 is mutated, mutant variants of Merlin are produced that do not act on the same functional level as wildtype Merlin isoforms. Mutated Merlin cannot effectively phosphorylate MST1/2 and/or LATS1/2, leading to YAP/TAZ interactions with TEADS1-4 in the nucleus and subsequent upregulation of cell proliferation-related gene expression. Although this interaction between mutant Merlin and Hippo signaling has yet to be explored in depth, failure of mutated Merlin to promote the Hippo “on” state in NF2-mutated cancerous tissue is thought to be a primary reason these tissues become cancerous in the first place. A loss of NF2 through chromosome 22 deletion similarly leads to the dysregulation of the Hippo signaling pathway, leading to cancer formation.

In the present review, we found that 8 of the 11 cancers studied were described as having mutations in NF2 and had some degree of Hippo signaling pathway modulation (Table 1). Not all the cancers studied had NF2 mutation implications to the same degree, with NF2 being a driver of mutation in some cancers, a non-driver of mutation in other cancers, and simply associated with mutations in other cancers. Cancers characterized as having driver mutations in NF2 were meningioma, schwannoma, ependymoma, mesothelioma, and renal cell carcinoma. Three of these tumors are sporadic CNS tumors that are often studied together. Although there is expanded research on NF2-mutated meningiomas and schwannomas, there seems to be a lack of studies on NF2-mutated ependymomas. For this reason, it is unclear whether the Hippo signaling pathway plays a role in the tumorigenesis of such ependymomas in a similar fashion to sporadic meningiomas and schwannomas. Mesothelioma is another cancer with NF2 driver mutations that has studies substantiating the involvement of Hippo signaling in its pathogenesis in humans. Renal cell carcinoma studies in humans and mouse models alike have provided evidence for Hippo signaling pathway involvement in the carcinogenesis of these NF2-mutated tumors.

Several of the cancers reviewed in the present study had non-driver mutations in NF2 that contributed to promoting tumorigenesis but were not essential mutations of these cancers. The cancers described as having non-driver NF2 mutations in this review are breast cancer, hepatocellular carcinoma, and melanoma. Although these cancers have been extensively studied, NF2 mutations are rarer in these cancers and have thus been less frequently reviewed. Numerous human and mouse studies have looked at NF2-mutations in breast cancers and have found that these tumors indirectly modulate Hippo signaling when simultaneously mutated with BRCA1 and BARD1. However, in TNBC tissue, NF2-mutant upregulation of the Hippo signaling pathway is thought to inhibit factors that contribute to worse prognoses. In hepatocellular carcinoma (HCC), there are conflicting studies in humans and mice regarding the involvement of Hippo signaling in NF2-mutant tumors. Human cell line studies provide evidence for Hippo signaling modulation in NF2-mutant HCC, whereas studies in mice negate Hippo involvement, providing evidence for the involvement of the EGFR pathway instead. More studies on NF2-mutant HCC must be done to further clarify the pathway’s involvement in the pathogenesis of these tumors. Similarly, in melanomas with NF2 mutations, studies in mice and humans provide conflicting evidence for Hippo signaling pathway involvement. While studies in mice support Hippo signaling modulation in NF2-mutant melanomas, studies in humans suggest that NF2-mutant melanomas are more integrally involved in altering the cGas STING signaling pathway.

Lastly, there were three cancers in this review that had studies describing NF2 mutations as simple associated mutations. Cancers associated with NF2 mutations include glioblastoma, prostate cancer, and thyroid cancer. Studies on NF2 involvement in these carcinomas are not extensive, and there remains much work to be done to understand how NF2 is involved in the carcinogenesis of these cancers. Prostate cancer and glioblastoma alike have studies in human cancer cell lines revealing NF2-mutations, but those studies do not describe any Hippo signaling pathway involvement. NF2-mutant glioblastomas have several theorized pathways, excluding Hippo signaling, while studies on NF2-mutant prostate cancer simply detail other tumor suppressors that are commonly associated with NF2 in those tumors. Interestingly, studies on thyroid cancer in mice and human cells alike have shown Hippo signaling pathway involvement in NF2-mutated tumors through the Ras pathway.

## 5. Conclusions

The Hippo signaling pathway is known to influence stem cell maintenance, differentiation, and regulation of the stem cell niche. Through modulation of the Hippo signaling pathway, NF2 mutations are directly linked to the regulation of stem cells [25]. Our review found that NF2-mutated tumors defined by this Hippo signaling pathway-mediated stem cell regulation include meningiomas, schwannomas, mesotheliomas, breast cancers, hepatocellular carcinomas, renal cell carcinomas, thyroid cancers, and melanomas. Although NF2 gene mutations define several cancers, targeted immunotherapies for specific NF2 mutations are currently unavailable. NF2 mutations in such tumors are known contributors to immunosuppression, although the specific mechanisms underlying these effects are unknown. Recently, several studies have linked NF2 mutations to the tumor immune microenvironment (TIME), which may provide a more targetable avenue for use of immunotherapeutics in NF2-mutated tumors [55].

Although NF2 mutations and their associated dysregulation of the Hippo pathway are characteristic of many cancers, it is important to discern NF2 mutants from a loss of NF2. Modulation in Hippo signaling pathway activity through NF2 is not contingent on a complete loss of NF2. Several studies of meningiomas and other cancers erroneously equated mutated NF2 with a complete loss of function. Several tumors are associated with chromosome 22 loss, resulting in an allelic complete loss of NF2 amongst other genes that reside on chromosome 22. However, further studies must be done to expand on the function of variant Merlin proteins resulting from mutated NF2. On a similar note, the nuanced functional effect of different NF2 mutations have yet to be explored. Some of the studies reviewed outline specific NF2 mutations which are commonly associated with a particular cancer type but do not detail how these different mutations relate to cancer formation. Future studies in these cancer fields must be conducted to fill this gap in the discernment of NF2 mutants’ effect on tumorigenesis as it relates to biological pathways like the Hippo signaling pathway.

## Figures and Tables

**Figure 1 cancers-17-00064-f001:**
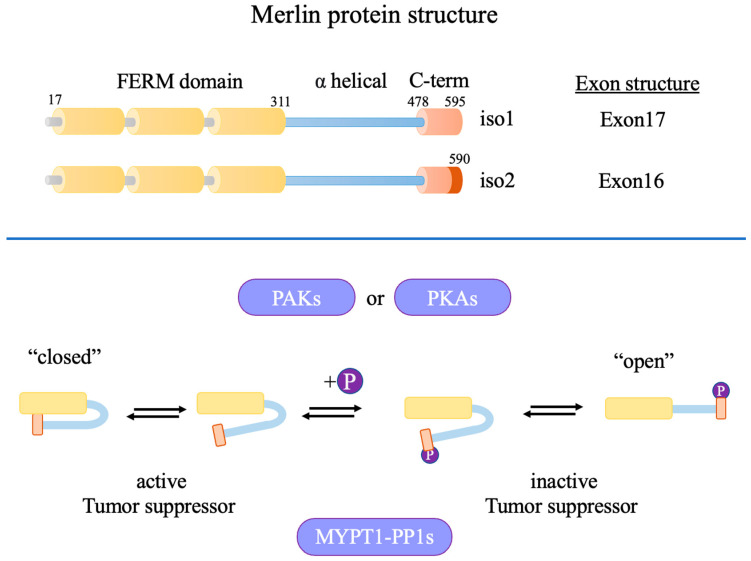
NF2/Merlin protein structure and conformational change upon phosphorylation.

**Figure 2 cancers-17-00064-f002:**
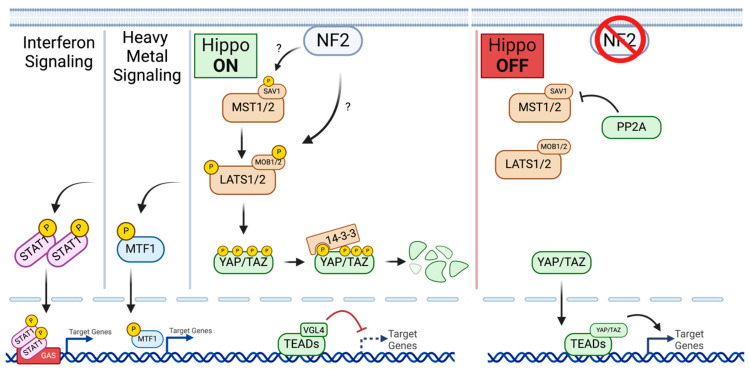
Diagram of Hippo Signaling Pathway “on” & “off” states. “?” indicates unknown specific pathway interaction. Created in BioRender.com.

**Table 1 cancers-17-00064-t001:** Various Cancers with NF2 mutations and Hippo Signaling Pathway involvement.

Cancer	NF2 Involvement	Species Studied	Hippo Signaling Pathway	Other Pathways Involved	NF2 Implication
Meningioma	Yes	*Homo sapiens*	Yes	CD44 regulated cell-cell inhibition, E-Cadherin membrane regulation	Driver mutation
Ependymoma	Yes	*Homo sapiens*	Unknown	Unknown	Driver mutation
Schwannoma	Yes	*Homo sapiens*	Yes	mTOR, PTEN in radioresistant tissue	Driver mutation
Mesothelioma	Yes	*Homo sapiens*	Yes	B-cell receptor signaling pathway	Driver mutation
Breast cancer	Yes	*Homo sapiens, Mus musculus*	Yes (via BRCA1 and BARD ubiquination)	CD24 and FSP1 in TNBC	Non-driver mutation
Hepatocellular Carcinoma	Yes	*Homo sapiens, Mus musculus*	Yes (in *H. sapiens* but not *M. musculus*)	EGFR pathway (in *M. musculus*)	Non-driver mutation
Prostate Cancer	Yes	*Homo sapiens*	Unknown	Other tumor suppressor mutations (OPCML, Cav-1, UGT2B17)	Associated mutation
Glioblastoma	Yes	*Homo sapiens*	Unknown	NOTCH1, EGFR, Rac1 pathways, HES1/CCND1 genes	Associated mutation
Thyroid Cancer	Yes	*Homo sapiens, Mus musculus*	Yes (in *M. musculus*)	Ras pathway coinvolved with Hippo signaling pathway	Associated mutation
Melanoma	Yes	*Homo sapiens, Mus musculus*	Yes (in *M. musculus*)	cGas STING signaling pathway	Non-driver mutation
Renal Cell Carcinoma	Yes	*Homo sapiens, Mus musculus*	Yes	Other tumor suppressor mutations (VHL, FH, cMET, SDH, SMARCB1)	Driver mutation

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
