# Peer review of "Role of NF2 Mutation in the Development of Eleven Different Cancers"

_cancers, 2024, doi:10.3390/cancers17010064_

Round 1
Reviewer 1 Report
Comments and Suggestions for Authors
Here, the authors propose a literature review on the NF2 gene and its implications in various cancers provides a comprehensive overview of the role of NF2 mutations in cancer development. The review is well written and should be of interest for the readership of the journal.
Overall, the review provides valuable insights into the role of NF2 mutations in cancer, highlighting both the progress made and the gaps that still need to be addressed.
They highlight how in some cancers, such as hepatocellular carcinoma and melanoma, there are conflicting studies regarding the involvement of the Hippo signaling pathway, indicating the need for further research.
Minor points:
While the authors discuss how NF2 acts as a driver mutation in some cancers, a non-driver mutation in others, and has simple associated mutations in other cancers, indicating a variable impact on cancer development, it would be interesting to discuss the co-occurence of NF2 mutations with other drivers, such as SMARCB1 or LZTR1 in nerve cancer (that are also located on the 22q region in cis of Nf2). Please cite: PMID: 30006736 that describes the 2/3 steps model and PMID: 28824165.
Additional evidence also functionally link Lats1/ 2 activity with these NF2 mutations as well as co-occuring mutations, see PMID: 38522513, PMID: 36148553 PMID: 37890861. It would be important to discuss the link between NF2, lats1/ 2 with MTF1 (PMID: 35027733) and STAT1 signaling (PMID: 35394840).
It would be also interesting to describe mutations that are mutually exclusive from NF2 alterations, such as CDK2A or TRAF7 in meningioma PMID: 36167400, PMID: 34215617.
Author Response
Here, the authors propose a literature review on the NF2 gene and its implications in various cancers provides a comprehensive overview of the role of NF2 mutations in cancer development. The review is well written and should be of interest for the readership of the journal. 
 
 
 
Overall, the review provides valuable insights into the role of NF2 mutations in cancer, highlighting both the progress made and the gaps that still need to be addressed.
They highlight how in some cancers, such as hepatocellular carcinoma and melanoma, there are conflicting studies regarding the involvement of the Hippo signaling pathway, indicating the need for further research. 
Minor points:
While the authors discuss how NF2 acts as a driver mutation in some cancers, a non-driver mutation in others, and has simple associated mutations in other cancers, indicating a variable impact on cancer development, it would be interesting to discuss the co-occurence of NF2 mutations with other drivers, such as SMARCB1 or LZTR1 in nerve cancer (that are also located on the 22q region in cis of Nf2).  Please cite: PMID: 30006736 that describes the 2/3 steps model and PMID: 28824165. 
Thank you for your feedback. We included a new paragraph within the introduction of the paper (lines 101 to 113) outlining the 4-hit/3-step model that details co-occurrence of NF2 mutations with SMARCB1 or LZTR1 in schwannomas.
“Within the q arm of chromosome 22 are adjacent genes to NF2 that are commonly co-mutated with NF2 in central nervous system tumors. Approximately 8-Mb away from NF2 and commonly involved in schwannoma tumorigenesis is the LZTR1 gene. According to the 4-hit/3-step model of schwannomagenesis, patients have a germline LZTR1 mutation that acts as the first hit, followed by loss of heterozygosity (LOH) as the second hit. LOH is secondary to a mitotic recombination event or large deletion that encompasses deletion of not only LZTR1, but also the neighboring NF2 allele. After this third mutational hit, a fourth hit occurs via an iatragenic NF2 mutation, leading to co-mutations of NF2 and LZTR1. The SMARCB1 gene is similar located nearby NF2 on chromosome 22q and is commonly co-mutated with NF2 following the 4-hit/3-step model [11]. Studies have shown evidence for additional somatic mutations in NF2 in schwannomas characterized by germline mutations in LZTR1 or SMARCB1 alike, further supporting evidence for the 4-hit/3-step model of schwannomagenesis [12].”
Additional evidence also functionally link Lats1/ 2 activity with these NF2 mutations as well as co-occuring mutations, see   PMID: 38522513, PMID: 36148553 PMID: 37890861. It would be important to discuss the link between NF2, lats1/ 2 with MTF1 (PMID: 35027733) and STAT1 signaling (PMID: 35394840).
We agree with your detailed comment and have since added to discuss the role of LATS1/2 in regulation of MTF1 and STAT1, and how NF2 mutations can alter these two signals. Moreover, we have included this important connection to the other signaling pathways in figure 2. The following was added in lines 217-231,
“Merlin is a known plasma membrane-associated protein required for the activation of hippo signaling cascade and thereby effecting the activity of LATS1/2. In addition to the effect that LATS1/2 has on downstream hippo signaling targets YAP and TAZ, LATS1/2 phosphorylation leads to the inhibition of MTF1, a transcription factor essential in heavy metal response. With NF2 mutations, LATS1/2 is downregulated, leading to lack of inhibition of MTF1, which results in upregulation of heavy metal response gene transcription and cellular protection. This was evident in NF2 KO cells that harbored resistance to Cd/Zn-induced toxicity [28]. LATS1/2 activity is also linked with regulation of interferon (IFNs), which have broad-spectrum antiviral activity. Upon LATS1/2 phosphorylation in the hippo signaling cascade, phospho-LATS1/2 directly promotes IFN-mediated antiproliferation. Additionally, activated LATS1/2 phosphorylates STAT1, inducing full IFN-I antiviral activity. With a deficiency in LATS1/2, it has been shown that in vivo IFN-I signaling is restricted, allowing for dysregulation of host antiviral immune response [29]. NF2 mutations lead to the inactivation of LATS1/2, thereby reducing the activity of STAT1 and attenuating IFN-I signaling activity.”
It would be also interesting to describe mutations that are mutually exclusive from NF2 alterations, such as CDKN2A or TRAF7 in meningioma PMID: 36167400, PMID: 34215617. 
Thank you for the comment and highlighting a very interesting point. Although mutations mutually exclusive to NF2 were mentioned briefly in line “Monosomy, NF2 mutations, and TRAF7 mutations seem to be the drivers of tumorigenesis in sporadic meningiomas, and as many as 60% of all sporadic meningiomas have loss of heterozygosity on chromosome 22 due to inactivating mutations of NF2 [15, 18, 19]”, we agree that it would be valuable to expand on other gene mutations. The following was added accordingly in lines 155-158,
“Meningiomas not defined by NF2 mutations can have deletions in CDKN2A/B located on chromosome 9p as well as TRAF7 mutations on chromosome 16p [20, 21]. Nearly a quarter of all meningiomas are characterized by TRAF7 loss, mutually exclusive to NF2 mutation [21].”

Reviewer 2 Report
Comments and Suggestions for Authors
The authors conducted a literature review on eleven different cancers with NF2 gene mutation involvement, summarizing the extent of association and specific biological pathways (e.g. the Hippo signaling pathway) thought to be affected by NF2 mutations. Some points should be noted as below1,
1) The title“Literature Review of NF2 gene and its implications in various cancers”, what about “its implications”,it seems no mention in Abstract section. It should be changed to something more appropriate?
2) In the first paragraph of the introduction, a description of cancer is provided (line 46-53), it has been (https://pubmed.ncbi.nlm.nih.gov/37056571/) proposed that cancer can be regarded as a pathological ecosystem in which the neoplastic cells interact with themself or with tumor microenvironment such as extracellular matrix, and immune cells. Such novel view might be helpful.
3) Line 152-159 about EMT, can the authors describe some histological evidence of EMT in human tumor tissues? Here are some examples,
â‘ Neoplastic spindle cells in nasopharyngeal carcinoma show features of epithelial-mesenchymal transition.Histopathology. 2012 Jul;61(1):113-22.
â‘¡Epithelial-mesenchymal transition in malignant mesothelioma. Mod Pathol. 2012. PMID: 21983934
4)Any immunotherapeutic application of NF2 gene involvement in human cancers?
5)How about NF2 gene regulating cancer stem cells in various cancers.
Author Response
1) The title “Literature Review of NF2 gene and its implications in various cancers”, what about “its implications”, it seems no mention in Abstract section. It should be changed to something more appropriate?
Thank you for this comment. We agree that the phrasing “implications” is vague and not directly addressed in the abstract. We therefore changed the title to “Role of NF2 mutation in the development of eleven different cancers” to more appropriately represent the paper (line 2).
2) In the first paragraph of the introduction, a description of cancer is provided (line 46-53, it has been (https://pubmed.ncbi.nlm.nih.gov/37056571/) proposed that cancer can be regarded as a pathological ecosystem in which the neoplastic cells interact with themself or with tumor microenvironment such as extracellular matrix, and immune cells. Such novel view might be helpful.
Thank you for this viewpoint. We agree that it is a more inclusive way of defining cancer and have since changed the sentence in question to (lines 49-52) and have cited the mentioned paper “In specific, cancer can be regarded as a pathological ecosystem that harbors tumor microenvironment factors, like extracellular matrix and immune cells, in which neoplastic cells interact with these factors and themselves [2]”
3) Line 152-159 about EMT, can the authors describe some histological evidence of EMT in human tumor tissues? Here are some examples,
①Neoplastic spindle cells in nasopharyngeal carcinoma show features of epithelial-mesenchymal transition. Histopathology. 2012 Jul;61(1):113-22.
â‘¡Epithelial-mesenchymal transition in malignant mesothelioma. Mod Pathol. 2012. PMID: 21983934
Thank you for the comment. Previously, it was mentioned in line 160-161 that “EMT has been associated with increased invasiveness of epithelial tumor cells”, yet examples were not specified. The following sentence has been added in lines 175-179 “Several studies have shown histological evidence of EMT in other epithelial cell cancers like nasopharyngeal carcinoma (NPC) and malignant mesothelioma (MM), which is thought to contribute to neoplastic spindle cell development in NPC and aggressive cancer phenotypes in MM [24, 25].”
4) Any immunotherapeutic application of NF2 gene involvement in human cancers?
Thank you for your comment on immunotherapeutics. Although targeted immunotherapies have not been discovered for specific NF2 mutations, this is a future direction for treatment of NF2-defined cancers. The following was included in our concluding remarks, “Although NF2 gene mutations define several cancers, targeted immunotherapies for specific NF2 mutations are currently unavailable. NF2 mutations in such tumors are known contributors to immunosuppression, although specific mechanisms underlying these effects are unknown. Recently, several studies have linked NF2 mutations with the tumor immune microenvironment (TIME), which may provide a more targetable avenue for use of immunotherapeutics in NF2-mutated tumors [56].” (lines 546-551)
5) How about NF2 gene regulating cancer stem cells in various
Thank you for your relevant comment. Although stem cell regulation in NF2-mutated cancers was indirectly discussed through Hippo signaling pathway involvement, we agree that it can be more explicitly stated. We included the following in our conclusion in lines 540-546, “The Hippo signaling pathway is known to influence stem cell maintenance, differentiation, and regulation of the stem cell niche. Through modulation of the Hippo signaling pathway, NF2 mutations are directly linked in the regulation of stem cells [26]. Our review found that NF2-mutated tumors defined by this Hippo signaling pathway-mediated stem cell regulation include meningiomas, schwannomas, mesotheliomas, breast cancers, hepatocellular carcinomas, renal cell carcinomas, thyroid cancers, and melanomas.”

Round 2
Reviewer 2 Report
Comments and Suggestions for Authors
The author has made very good revisions in accordance with the relevant opinion.